# Using the CERES-Maize Model to Simulate Crop Yield in a Long-Term Field Experiment in Hungary

Annabella Zelenák [1], Atala Szabó [1], János Nagy [1] and Anikó Nyéki [2,*]

[1] Institute of Land Use, Engineering and Precision Farming Technology, Faculty of Agricultural and Food Sciences and Environmental Management, University of Debrecen, 138 Böszörményi St., 4032 Debrecen, Hungary; zelenak@agr.unideb.hu (A.Z.); szabo.atala@agr.unideb.hu (A.S.); nagyjanos@agr.unideb.hu (J.N.)

[2] Department of Biosystems and Food Engineering, Széchenyi István University, 2. Vár Square, 9200 Mosonmagyaróvár, Hungary

* Correspondence: nyeki.aniko@sze.hu; Tel.: +36-307204606

**Abstract:** Precision crop production requires accurate yield prediction and nitrogen management. Crop simulation models may assist in exploring alternative management systems for optimizing water, nutrient and microelements use efficiencies, increasing maize yields. Our objectives were: (i) to access the ability of the CERES-Maize model for predicting yields in long-term experiments in Hungary; (ii) to use the model to assess the effects of different nutrient management (different nitrogen rates—0, 30, 60, 90, 120, and 150 kg ha$^{-1}$). A long-term experiment conducted in Látókép (Hungary) with various N-fertilizer applications allowed us to predict maize yields under different conditions. The aim of the research is to explore and quantify the effects of ecological, biological, and agronomic factors affecting plant production, as well as to conduct basic science studies on stress factors on plant populations, which are made possible by the 30-year database of long-term experiments and the high level of instrumentation. The model was calibrated with data from a long-term experiment field trial. The purpose of this evaluation was to investigate how the CERES-Maize model simulated the effects of different N treatments in long-term field experiments. Sushi hybrid's yields increased with elevated N concentrations. The observed yield ranged from 5016 to 14,920 kg ha$^{-1}$ during the 2016–2020 growing season. The range of simulated data of maize yield was between 6671 and 13,136 kg ha$^{-1}$. The highest yield was obtained at the 150 kg ha$^{-1}$ dose in each year studied. In several cases, the DSSAT-CERES Maize model accurately predicted yields, but it was sensitive to seasonal effects and estimated yields inaccurately. Based on the obtained results, the variance analysis significantly affected the year (2016–2020) and nitrogen doses. N fertilizer made a significant difference on yield, but the combination of both predicted and actual yield data did not show any significance.

**Keywords:** long-term experiments; maize yield prediction; CERES-Maize model; sustainable crop production

## 1. Introduction

Crop production is currently facing the challenge of meeting the increasing demand by using less fertilizers, water, and pesticides, while ensuring safety of food, including the presence of appropriate microelements. In order to clarify yield predictions, crop simulation models consider several factors, and they can also contribute to more precise, site-specific crop production [1]. Thus, it is crucial to integrate agronomy and decision support systems. Databases of long-term experiments, linking with the possibilities provided by plant physiological models, can create an integrated system in agricultural research, which can play an important role in mapping the hypotheses of the yield gap. The per-

formed field experiment has been pivotal in assessing the effects of single or multiple factors on crop productivity, because the crop yield prediction is based on soil, meteorological, crop, and environmental variables. The purpose of this study is to evaluate the simulation model, based on big data and site-specific measurements. Through this, we can improve the decision support system of precision agriculture. Maize (*Zea mays* L.) is the most important crop in Hungary [2]. Although, sustainability of high crop yield under intensive cultivation is possible only through the use of water, and the use of adequate chemical fertilizer, i.e., microelements.

The dataset of the Látókép long-term experiment (2016–2020) was used for the analysis of this study. By testing the model, the yield of maize was analyzed:

- Growth and yield of maize hybrids in a context of environmental conditions;
- The initial parameters for the model run were established (soil chemical variables, soil physical properties, soil mechanical structure, soil moisture, etc.);
- Phenological and growth characteristics of individual maize hybrids.

*Use of the DSSAT Software under Precision and Experimental Conditions*

In addition to comparing models, DSSAT is considered to be suitable for demonstrating the effect of phenological and N-fertilizer, but it is less suitable for expressing water stress or soil moisture. Modelling has been the subject of numerous publications and studies, and continuous calibration and validation (temporal and spatial scaling) have been shown to be essential to achieve the goals of sustainable crop production. They also became important because of new scientific directions and hypotheses [3]. Based on the database of the Látókép (Hungary) long-term experiment, maize hybrids were analyzed for the effects of irrigation, soil tillage, and crop number [4,5], taking into account different season effects. The sensitivity of the CERES-Maize model extends to extreme meteorological years and vegetation periods. The model over- or underestimates the grain yield in rainy and drier years. It overpredicted the yields systematically in extreme rainy years [6,7] and under irrigated conditions [8]. For some treatments, the model overestimated corn grain yield and underestimated total N uptake as well as underestimated total leached nitrogen and soil moisture, which has an effect on yield [9]. Grain yield increased with increasing nitrogen content; however, the model underpredicted grain yield with control treatment [10]. Li et al. [11] observed a significant difference in two years, which were extremely dry. There is also a similar experience with Liu et al. [6] that, in a low-rainy year, the maize yields were undersimulated by the model. In the study [5], the average percentage error of maize predictions for the run environment ranged from 4.8% to 46.6%, with differences of 471 to 2407 kg/ha. Accurate, reliable yield estimates could be given by measurements taken during the vegetation period of the crop, as the inaccuracy of the prediction at sowing can reach 50% [12]. According to Quiring and Legates [5], the model is partially sensitive to row spacing, seed, sowing depth, sowing, and harvesting time, hybrid, soil type and soil moisture, as well as temperature and global radiation. Specifically, the authors draw attention to the effect of the relationship between soil and precipitation, and the time that can be extremely sensitive to yield development. Li et al. [11] used the model to analyze a small plot experiment.

In addition to yield estimation, DSSAT can to run various dynamics, such as soil dynamics, simulation of the nitrogen cycle, and monitoring of changes in soil organic matter content [13]. The crop simulation program contributes to decision making for environmental risks, this model is applied to the evolution of the climatic conditions [14]. Provides crop simulation models for management decision making, risk management, and evaluation [15]. The CERES-Maize model was applied under farm conditions [16], and embedded in the Apollo [17] decision support system. A 20.25 ha experimental parcel [18] was divided into nearly 100 management zones, based on which the current and future corn yield was validated in the Apollo framework. Based on his studies, he found that later larger treatment units also could be effective as yields show a smaller spatial distribution.

Paz et al. [19] tested the optimum (141–160 kg/ha) application of N (between 60 and 220 kg/ha) on ~500 m² units of a 16 ha parcel. DSSAT and APSIM models were examined, and the most notable difference in treatment units was 6 tons based on a comparison of the three-year model estimates. Salmerón et al. [20] simulations in the La Violada watershed did not adequately estimate yield loss under high yield conditions under reduced nitrogen. Furthermore, under optimized water and nitrogen management N leaching (44–98 kg N ha⁻¹ yr⁻¹) would still be high. Zhu et al. [21] emphasizes that without the adaptation of precision crop cultivation techniques, or in the absence of these data, the development of models for plant physiology and agro-ecosystems and the use of newer model generations will fail. However, there is little information on modelling the combined effects of water and N limitations on water productivity responses of maize to irrigation.

## 2. Materials and Methods

### 2.1. Experimental Site and Treatments

The Látókép Crop Production Experiment Site was established in 1983. The area of the Látókép experiment is 190 ha, most of which, i.e., a 125 ha site, can be irrigated. The experiment site is located in eastern Hungary in the Hajdúság region (47°33′27″ N, 21°26′52″ E.) (Figure 1.) The site is relatively isolated, which provided excellent conditions for establishing long-term experiments. Over the past 38 years, the experiment has remained unchanged in terms of location, nutrient replenishment rate, soil tillage, and agricultural elements. The field experiment is arranged in a randomized complete block design with 360 blocks (including 6 treatments, 15 hybrids in four replications). Size of one repetition: 1260 m², for fertilizer plots: 210 m². In this study we examined 1 hybrid including 6 treatments in four replications. There were 24 blocks in a year

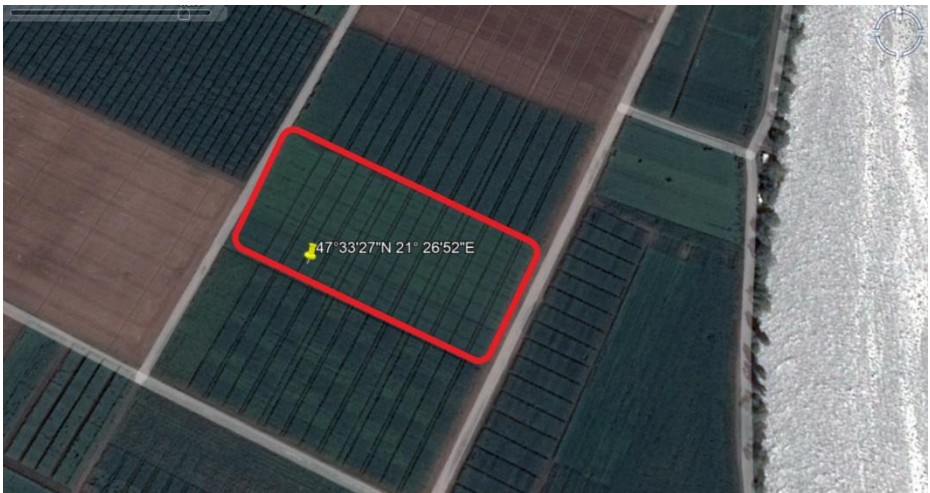

**Figure 1.** The experimental design in Látókép.

The recommended fertilizer doses were 0: 0 kg ha⁻¹ N, 1: 30 kg ha⁻¹ N, 2: 60 kg ha⁻¹ N, 3: 90 kg ha⁻¹ N, 4: 120 kg ha⁻¹ N, and 5: 150 kg ha⁻¹ N in the experimental treatments. The application rates of chemical fertilizers are described in Table 1, based on a soil analysis recommendation. In the experiment, 30% of the total nitrogen dose and 100% of the phosphorus and potassium doses were applied at the beginning, before plowing, and 70% of the nitrogen dose was applied as top-dressing in April.

**Table 1.** Fertilizer input of different treatments in our experiment.

| Fertilizer Amount | N | $P_2O_5$ | $K_2O$ | Sum |
|---|---|---|---|---|
| 0 | - | - | - | - |
| 1 | 30 | 23 | 27 | 80 |
| 2 | 60 | 46 | 54 | 160 |
| 3 | 90 | 69 | 81 | 240 |
| 4 | 120 | 92 | 108 | 320 |
| 5 | 150 | 115 | 135 | 400 |

*2.2. Soil Data*

The long-term experiment was set up to collect data for calibrating and validating the CERES-Maize model. It was conducted on silty clay loam soil as classified in USDA [22] (Table 2.). The collected soil samples were analyzed for texture, bulk density, pH, organic matter, total nitrogen, potassium (K), and phosphorus (P). The soil of the experiment site is calcareous chernozem formed on the Hajdúság loess ridge with 80–90 cm depth top soil, and the organic matter content is around 2.7%. The soil has a pH of 6.6 (slightly acidic). In terms of its physical variety, it is a clayey loam, with Arany's plasticity index number of 44 in 2017 and 2020. The soil input dataset was created by measuring soil properties such as soil texture, soil bulk density, pH, organic carbon, total N, and available phosphorus and nitrogen.

**Table 2.** Soil physical properties in the experimental site (2020).

| Depth (cm) | Sand | | Silt | | | | | Clay |
|---|---|---|---|---|---|---|---|---|
| cm | 2–0.25 | 0.25–0.05 | 0.05–0.02 | 0.02–0.01 | 0.01–0.005 | 0.005–0.002 | <0.002 | |
| 0–20 | 0.08 | 8.24 | 34.81 | 12.78 | 8.55 | 7.47 | 28.07 | |
| 20–45 | 0.04 | 8.6 | 32.72 | 15.34 | 8.05 | 7.34 | 27.91 | |
| 45–65 | 0.04 | 10.39 | 32.43 | 15.51 | 7.8 | 8.81 | 25.02 | |
| 65–95 | 0.24 | 10.15 | 29.11 | 14.88 | 7.87 | 10.12 | 27.63 | |
| 95–105 | 0.2 | 8.36 | 32.82 | 15.93 | 7.79 | 8.2 | 26.7 | |
| 105–140 | 0.32 | 11.5 | 34.08 | 15.7 | 8.12 | 7.28 | 23 | |

*2.3. Climatic Conditions of the Experimental Site (2016–2020)*

The experiment data were obtained from Centre for Agricultural Sciences, Institute of Crop Sciences, at Látókép, and the daily weather data from the Meteorological Observatory Debrecen of the National Meteorological Service. It was performed on the basis of the data of an automatic weather station set up next to the experimental plots [23] (Tables 3 and 4). The station provided the daily radiation, precipitation, wind speed, as well as minimum and maximum temperature data. This area has a typical continental climate with one growing season for maize production: from April to October. The mean annual rainfall is around 600 mm, the distribution of which causes strong atmospheric drought at times, resulting in low maize yield [24].

**Table 3.** Mean monthly air temperature (°C) in the growing season of maize at Debrecen-Látókép (2016–2020).

| | 2016 | 2017 | 2018 | 2019 | 2020 |
|---|---|---|---|---|---|
| April (1) | 13.3 (+2.1) | 10.7 (−0.5) | 16.0 (+4.8) | 12.4 (+1.2) | 10.8 (−0.4) |
| May (2) | 16.5 (−0.1) | 17.2 (+0.6) | 19.7 (+3.1) | 14.1 (−2.5) | 14 (−2.6) |
| June (3) | 21.1 (+1.8) | 22.2 (+2.9) | 20.2 (+0.9) | 22.8 (+3.5) | 19.6 (+0.3) |
| July (4) | 22.3 (+1.0) | 22.3 (+1.0) | 21.7 (+0.4) | 21.1 (−0.2) | 20.9 (−0.4) |
| August (5) | 20.8 (0) | 23.2 (+2.4) | 23.2 (+2.4) | 23.1 (+2.3) | 22.6 (+1.8) |
| September (6) | 17.6 (+1.6) | 16.4 (+0.4) | 17.1 (+1.1) | 17.1 (+1.1) | 17.9 (+1.9) |

| | | | | | |
|---|---|---|---|---|---|
| October (7) | 9.7 (−0.9) | 10.8 (+0.2) | 12.3 (+1.7) | 12.6 (+2.0) | 11.7 (+1.1) |
| Summer period (IV–IX.) (8) | 18.6 (+1,1) | 18.7 (+1.2) | 19.7 (+2.2) | 18.4 (+0.9) | 17.6 (+0.1) |
| Winter period (X–III.) (9) | 3.9 (−0.3) | 4.1 (−0.1) | 4.1 (−0.1) | 4.4 (+0.2) | 4.6 (+0.4) |

Note: Differences (in °C) from the climatic normal values of 1981–2010 are shown in brackets; (1) April, (2) May, (3) June, (4) July, (5) August, (6) September, (7) October, (8) Summer period (April–September), (9) Winter period (October–March).

**Table 4.** Monthly sum of precipitation (mm) in the growing season of maize at Debrecen-Látókép (2016–2020).

| | 2016 | 2017 | 2018 | 2019 | 2020 |
|---|---|---|---|---|---|
| April (1) | 16 (−37) | 51 (−2) | 37 (−16) | 33 (−20) | 17 (−36) |
| May (2) | 68 (+4) | 27 (−37) | 57 (−7) | 76 (+12) | 45 (−19) |
| June (3) | 146 (+80) | 67 (+1) | 64 (−2) | 32 (−34) | 119 (+53) |
| July (4) | 87 (+21) | 73 (+7) | 55 (−11) | 99 (+33) | 188 (+122) |
| August (5) | 72 (+23) | 61 (+12) | 92 (+43) | 15 (−34) | 70 (+21) |
| September (6) | 64 (+16) | 76 (+28) | 14 (−34) | 35 (−13) | 44 (+4) |
| October (7) | 98 (+60) | 38 (0) | 9 (−29) | 22 (−16) | 79 (+41) |
| Summer period (IV–IX.) (8) | 453 (+107) | 354 (+8) | 318 (−28) | 290 (−56) | 483 (+137) |

Note: Differences (in mm) from the climatic normal values of 1981–2010 are shown in brackets; (1) April, (2) May, (3) June, (4) July, (5) August, (6) September, (7) October, (8) Summer period (April–September).

After the winter period of 2016, the considerably dry and warm April had a positive effect. In April, a total amount of 16 mm of rain fell on several occasions. The remaining part of the growing season was characterized by high rainfall and above-average temperatures. The ideal conditions were provided for maize growth and its yield. Precipitation was above average in each month. The total precipitation for the summer semester is 453 mm, of which the values of 146 mm in June and 87 mm in July should be highlighted. The temperature was mostly above average, but there was no long and extremely warm period. The months of September (+1.6 °C) and June (+1.8 °C) showed positive temperature anomalies. The average temperature in August conformed to the multi-year average.

Regarding the summer months (2017), precipitation (354 mm) is essentially the same as the multi-year average (+8 mm), and it was well balanced in terms of monthly precipitation. There was a significant difference from the multiple-year average in May (−37 mm) and September (+28 mm). Significant positive temperature anomalies occurred in June (+2.9 °C), July (+1.0 °C), and August (+2.4 °C).

In the year 2018, from the beginning of April, the nature of weather changed fundamentally and permanently, which is also well reflected in the monthly data. The growing season started with a very warm April, with an average temperature of 16.0 °C, almost 5 °C above the average value. Sunny, warm weather continued in May, again resulting in a record high average temperature (19.7 °C). These two significantly warm months contributed favorably to the emergence and initial development of maize.

The positive temperature anomaly in April 2019 (+1.2 °C) was followed by a significantly negative anomaly in May (−2.5 °C). The development of maize was slow, but the water supply was optimal. In the summer months, the temperatures in June (+3.5 °C) and August (+2.3 °C) were well above average. The positive anomaly continued into the fall. During the growing season, April (−20mm), June (−34 mm), and July (−34 mm) had significantly below average rainfall. The 99 mm rainfall in July was favorable during the silking and grain filling phase of maize.

In the initial growing season in 2020, the significant negative temperature anomaly (−0.4 °C in April; −2.6 °C in May) was associated with low precipitation. Spring precipitation was followed by particularly high monthly meteorological values during the three summer months. This weather negatively affected the development of the sown maize

and its emergence. During the summer months, there was a significant surplus of precipitation compared to the multiple-year average (June +53 mm, July +122 mm, August +21 mm). This rainy weather continued into the fall months. The difference in the average monthly temperatures from the average continued even in August (+1.8 °C), September (+1.9 °C), and October (+1.1 °C) months.

*2.4. Model Calibration and Evaluation/Data Requirements for Calibrating and Validating the Ceres-Maize Model*

The CSM-CERES Maize model (v.4.7) is a deterministic model to simulate crop growth and development on a daily basis [25]. The CERES-Maize model simulates conversion processes of soil water, carbon, and nitrogen balances and predicts maize yield and N uptake, as well as water use efficiency. Daily records of minimum and maximum temperature, total rainfall, and solar radiation are required for the model. The Weatherman utility also needs information from the weather station. Soil data tool (SBuild, Version 4.7.5, DSSAT Foundation, Gainesville, Florida, USA) was used to adapt site coordinates, soil profile, and classification. Measured soil characteristics were used to calculate soil physical and chemical parameters that are needed to run the model for yield prediction (Tables 5 and A1). For simulation options, initial conditions were reported for each year and location (hybrids). The Priestly–Taylor (Ritchie method) was selected for simulating evapotranspiration and the Soil Conservation Service method for infiltration. The Ritchie Water Balance model was set for soil evaporation. Photosynthesis was configured, while phosphorus and potassium were not modelled in all runs.

**Table 5.** The physical and chemical parameters of the soil in the experimental area in 2020.

| | | | | | 2020 | | | | |
|---|---|---|---|---|---|---|---|---|---|
| Layer Depth, cm | Organic Carbon % | Total Nitrogen % | pH in Water | Lower Limit, $cm^3$ $cm^{-3}$ | Drained Upper Limit, $cm^3$ $cm^{-3}$ | Saturated Water Holding Capacity, $cm^3$ $cm^{-3}$ | Bulk Density g/$cm^3$ | Sat. Hydraulic Conduct, cm/h | Root Growth Factor, 0.0 to 1.0 |
| 5 | 1.39 | 0.13 | 7.3 | 0.204 | 0.414 | 0.489 | 1.26 | 0.15 | 1.000 |
| 10 | 1.39 | 0.13 | 7.3 | 0.204 | 0.414 | 0.489 | 1.26 | 0.15 | 1.000 |
| 15 | 1.45 | 0.14 | 7.3 | 0.206 | 0.417 | 0.488 | 1.26 | 0.15 | 1.000 |
| 20 | 1.45 | 0.14 | 7.3 | 0.206 | 0.417 | 0.488 | 1.26 | 0.15 | 1.000 |
| 25 | 1.39 | 0.13 | 7.2 | 0.203 | 0.412 | 0.489 | 1.26 | 0.15 | 0.638 |
| 30 | 1.39 | 0.13 | 7.2 | 0.203 | 0.412 | 0.489 | 1.26 | 0.15 | 0.577 |
| 35 | 1.59 | 0.15 | 7.2 | 0.209 | 0.423 | 0.495 | 1.24 | 0.15 | 0.522 |
| 40 | 1.59 | 0.15 | 7.2 | 0.209 | 0.423 | 0.486 | 1.24 | 0.15 | 0.472 |
| 45 | 0.27 | 0.12 | 7.2 | 0.2 | 0.406 | 0.493 | 1.27 | 0.15 | 0.427 |
| 50 | 0.27 | 0.12 | 7.2 | 0.186 | 0.392 | 0.484 | 1.25 | 0.68 | 0.387 |
| 55 | 0.95 | 0.09 | 7.2 | 0.177 | 0.376 | 0.484 | 1.28 | 0.68 | 0.35 |
| 60 | 0.95 | 0.09 | 7.2 | 0.177 | 0.376 | 0.484 | 1.28 | 0.68 | 0.317 |
| 65 | 0.81 | 0.08 | 8.0 | 0.173 | 0.369 | 0.478 | 1.3 | 0.68 | 0.287 |
| 70 | 0.81 | 0.08 | 8.0 | 0.186 | 0.378 | 0.478 | 1.3 | 0.15 | 0.259 |
| 75 | 0.75 | 0.07 | 8.0 | 0.184 | 0.375 | 0.479 | 1.3 | 0.15 | 0.235 |
| 80 | 0.75 | 0.07 | 8.0 | 0.184 | 0.375 | 0.479 | 1.3 | 0.15 | 0.212 |
| 85 | 0.92 | 0.09 | 8.4 | 0.189 | 0.384 | 0.481 | 1.29 | 0.15 | 0.192 |
| 90 | 0.92 | 0.09 | 8.4 | 0.189 | 0.384 | 0.481 | 1.29 | 0.15 | 0.174 |

*2.5. Weather Data*

Daily records of solar radiation amount (SRAD), maximum temperature ($T_{max}$), minimum temperature ($T_{min}$), and precipitation (RAIN), wind speed, and the relative humidity (RHUM) are required for the CERES-Maize model. According to Banda (2005) [26], the most important factors are the intensity and distribution of precipitation during the growing season, which greatly influences the development of maize yield.

*2.6. Examined Hybrid Parameters*

In the field experiment (11 measuring points), hybrids were characterized in the vegetation period of 2018 on the basis of Hanway's scale, and values were used to determine genetic parameters [27]. To calibrate the genetic coefficients of the maize cultivar, dates of emergence, silking and physiological maturity, biomass, grain yield, and leaf area index were used. These phenological parameters include thermal time from seedling emergence to the end of the juvenile phase (P1), photoperiod-sensitivity (P2), thermal time from silking to physiological maturity above base temperature of 8 °C (P5), potential kernel number (G2), potential grain filling rate (G3), and interval in degree-days between successive leaf tip appearance (PHINT) (Table 6). In this study, we selected one hybrid Sushi (FAO 340). Crop development was assessed by observing the phenology of different maize varieties and recording the daily sum of heat required to reach each phenological phase (Hanway, 1963) [28]. Maize is a heat-demanding crop, but temperatures higher than 30 °C are not taken into account in the heat sum calculation. The total heat demand of the hybrids during the growing season is 1100–1400 °C [24]. Hanway (1963) [28] determined the growth stages before silking based on the number of leaves, the subsequent stages defined on kernel development, the growing season of maize was divided into eleven growth stages [24].

**Table 6.** Genetic coefficients for the Sushi hybrid.

| Hybrid | P1 | P2 | P5 | G2 | G3 | PHINT |
|--------|-----|-------|-----|-----|-----|-------|
| Sushi | 118 | 0.500 | 926 | 830 | 7.1 | 42 |

The Sushi hybrid was sown on 19 April 2016, 25 April 2017, 24 April 2018, 16 April 2019, and 17 April 2020. The harvesting dates were in 14 October 2016, 12 October 2017, 19 September 2018, 16 October 2019, and 24 October 2020. The sowing machine was set by 76 cm at intervals 18 cm in row with 70.000 ha$^{-1}$ seedlings (Table 7).

**Table 7.** Sowing and harvest dates between 2016 and 2020 years.

| Years | Sowing Date | Harvest Date |
|-------|-------------|--------------|
| 2016 | 19 April | 14 October |
| 2017 | 25 April | 12 October |
| 2018 | 24 April | 19 September |
| 2019 | 16 April | 16 October |
| 2020 | 17 April | 24 October |

Initial conditions were based on those reported for each year and location (hybrids). The Priestly–Taylor (Ritchie method) was selected for simulating evapotranspiration and the Soil Conservation Service method for infiltration. The Ritchie Water Balance model was set for soil evaporation. Photosynthesis was configured and Phosphorus and Potassium were not modelled in all runs. In addition to the agrotechnological applications performed during the long-term experiment (tillage, sowing, fertilizer application, method, and dates of harvest), additional information was used to build the model. The objective of this study is to analyze the effect of fertilizer doses on yields and the differences between seasons.

*2.7. Statistical Analysis*

Statistical assessment to judge the accuracy of CERES-Maize outputs included the root mean square error (RMSE), normalized-RMSE (nRMSE). These indicators were measured as Yang and Huffman (2004) [29]:

$$RMSE = \sqrt{\frac{\sum_{i=1}^{n}(m_i - s_i)^2}{n}}$$

$$n - RMSE = \frac{RMSE}{\bar{m}}$$

where $n$ is the number of measured dataset, $s_i$ is simulated data, $m_i$ is measured data, and $\bar{m}$ is the mean of the measured data. In addition, because $n$-RMSE is unbounded, and it is unstable if $\bar{m}$ or n approaches zero, an index of agreement (d) statistic was used in this study [30].

In this study, the univariate statistical method was used (regression analysis and analysis of variance) to analyze hybrids and NPK treatment interactions. Regression examines the trend in the relations and describes the mode of the relation with a certain function, i.e., it quantifies the causal relations. The regression coefficient gives the average change in the "explanatory" variable per unit change in the "response" variable,

$$Y = \beta_0 + \beta_1 X,$$

where $\beta_1$ represents the regression coefficient. Parameter $\beta_0$ can usually only be interpreted mathematically if the variable X is set to 0, then $\beta_0$ is the estimate given 0 in X. The linear correlation coefficient is known as the coefficient of determination ($R^2$), and it shows the percentage of the variance of the response variable is explained by the factor variable, explain its reliability.

Comparing two or more groups observed by ANOVA (variance of analysis) shows if there are significant differences. The value of $R^2$ is between 0 and 1, and it expresses a percentage of the strength of the relations between the variables. The assay performs an F-test. The value can be used to determine if the test is statistically significant. The program determines the p-value from the F-value, which determines whether the treatments have produced significant results. If the results are significant, the model is predictably valid. All analyses were performed using Minitab.

**3. Results and Discussion**

The weather variations from year to year in this study were significant. Examining the five years, 2016 and 2020 can be considered significantly rainy years. In 2018 and 2019, high average temperature values were associated with drought [23,31]. The total rainfall in the experimental years was 817 mm (2016), 641 mm (2017), 552 mm (2018), 479 mm (2019), and 708 mm (2020).

*3.1. CERES-Maize Simulation Results*

The CERES-Maize model was used to analyze the N-fertilization experiments conducted in Látókép during the given growing seasons (2016–2020). The hybrid Sushi (FAO 340) that we tested was calibrated from preliminary field and crop phenological measurements. From the long-term experiment field measurement results and according to the agrotechnical elements for the given year, simulation settings were determined. The performance of the CERES-Maize model was evaluated by comparison between simulated and observed grain yield under different N treatments (Table 8). The observed yield ranged from 5016 to 14,920 kg ha$^{-1}$ during the 2016–2020 growing season, respectively. Higher yields (at 150 kg ha$^{-1}$ N dose) were measured in the rainy years: 13,858 kg ha$^{-1}$ (2016), 13,400 kg ha$^{-1}$ (2020), with the exception of the average rainfall year 2018 (14,920 kg ha$^{-1}$). The simulated data of maize yield ranged between 6671–13,136 kg ha$^{-1}$. Simulated

and observed maize yield results are similar to the results obtained by Bao et al. (2017) [32], and in other research works [9–12]. The aim of this evaluation was to investigate how the CERES-Maize model simulated the effects of different N treatments on the observed yield data in long-term experiments, in Hungarian conditions. In both years (2016 and 2017), the maximum yield of the Sushi hybrid was achieved at 150 kg ha$^{-1}$ N [33].

**Table 8.** Measured and simulated yields of Sushi maize hybrid (2016–2020).

| Year | N Rate (kg N ha$^{-1}$) | Grain Yield (kg ha$^{-1}$) | |
|------|------|------|------|
| | | **Measured** | **Simulated** |
| 2016 | 0 | 8657 | 8838 |
| | 30 | 11,036 | 10,010 |
| | 60 | 12,318 | 11,010 |
| | 90 | 12,773 | 11,792 |
| | 120 | 13,467 | 12,624 |
| | 150 | 13,858 | 13,136 |
| 2017 | 0 | 5016 | 8344 |
| | 30 | 6629 | 8638 |
| | 60 | 8627 | 9095 |
| | 90 | 9652 | 9438 |
| | 120 | 11,011 | 9866 |
| | 150 | 11,688 | 10,004 |
| 2018 | 0 | 6995 | 6671 |
| | 30 | 9980 | 6984 |
| | 60 | 11,540 | 7338 |
| | 90 | 12,030 | 7525 |
| | 120 | 14,640 | 7595 |
| | 150 | 14,920 | 7592 |
| 2019 | 0 | 7200 | 7740 |
| | 30 | 9920 | 8639 |
| | 60 | 9940 | 9181 |
| | 90 | 9780 | 9269 |
| | 120 | 10,240 | 9387 |
| | 150 | 10,860 | 9392 |
| 2020 | 0 | 5488 | 10,520 |
| | 30 | 7999 | 11,186 |
| | 60 | 8629 | 11,864 |
| | 90 | 10,259 | 12,056 |
| | 120 | 11,757 | 12,056 |
| | 150 | 13,400 | 12,056 |

Based on the obtained results, the model most accurately predicted the yield of our test plant in 2016 and 2019. The agrotechnical settings were used according to the given year and the soil test results were also different only in 2020 due to the new sampling tests. Figures 2–6 show the comparisons of the measured and estimated yields by the simulation model. The good fit between predicted and measured yield data showed that the model could be relevant to simulate the performance of yield for the dissimilar N treatments. Maize grain yield is significantly affected by fertilization (N). In the examined years, the Sushi hybrid yields increased with increasing doses of N. In 2016, except for the N0 dose, the model underestimated the yield in all cases. Due to the year with favorable rainfall, the high yields are also reflected in the simulation results. The measured yield data were determined with $R^2 = 0.88$ ($R^2$ = coefficient of determination) and the simulated results

with $R^2 = 0.98$. One unit increase in nitrogen dose increased the yield by 32.14 kg/ha for the measured data, while it was 28.68 kg/ha for the simulated data (Figure 2).

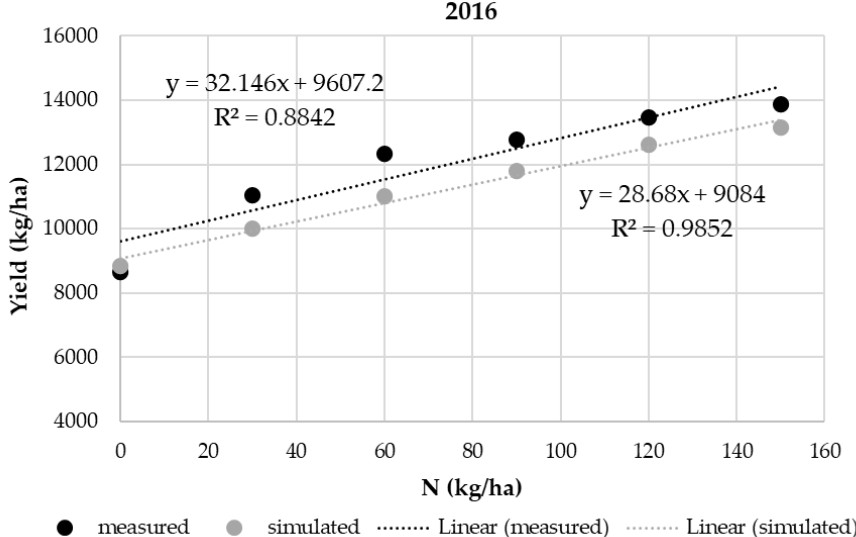

**Figure 2.** Comparisons between simulated and measured maize grain yield (2016).

An increase in the nitrogen dose per unit increases the yield by 45.26 kg/ha for the measured data, while it was 11.74 kg/ha for the simulated data. Up to the N60 dose, the model estimated yields at the top and then at the bottom (Figure 3) In 2018, the simulated results ($R^2 = 0.87$) followed the increase in yield in parallel with the increase in the fertilizer dose. However, the values lagged far behind the measured results ($R^2 = 0.94$). An increase in the nitrogen dose per unit increases the yield by 51.51 kg/ha for the measured data, while it was 6.30 kg/ha for the simulated data (Figure 4)

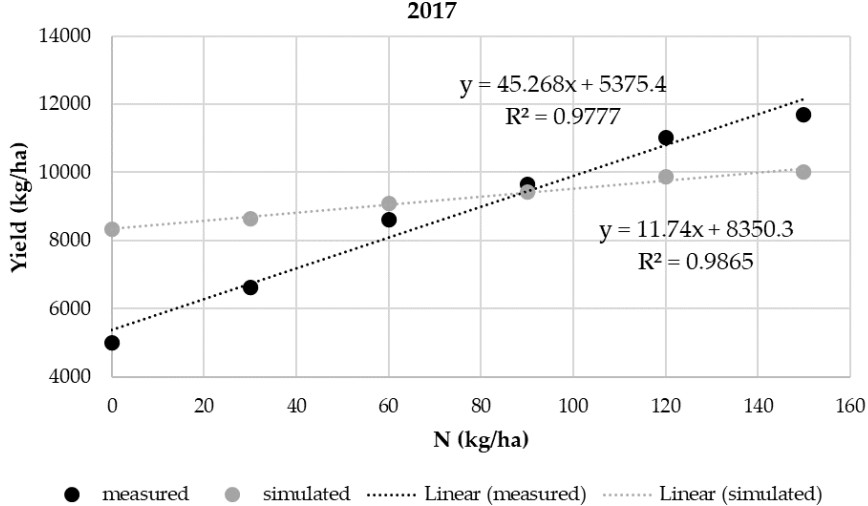

**Figure 3.** Comparisons between simulated and measured maize grain yield (2017).

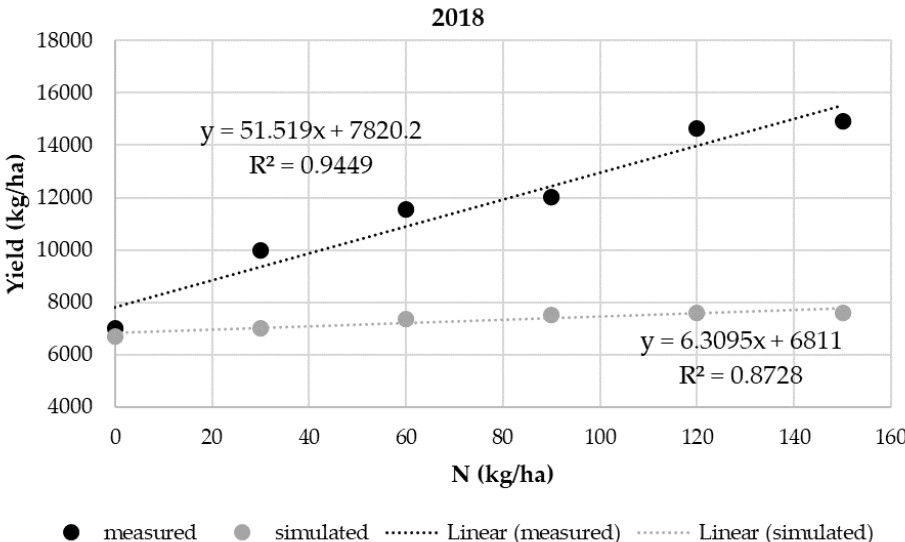

**Figure 4.** Comparisons between simulated and measured maize grain yield (2018).

In 2019, the measured yields were also below the average and the highest yield was 10 t/ha. An increase in the nitrogen dose per unit increases the yield by 18.19 kg/ha for the measured data, while it was 10.08 kg/ha for the simulated data. This trend was followed by the simulation yield, but underestimated the yields in all cases except for the N0 dose. The measured results were determined with $R^2 = 0.65$ and the estimated yields with $R^2 = 0.76$ (Figure 5) In the year 2020, the measured yield values are $R^2 = 0.98$ and the simulated $R^2 = 0.77$. An increase in the nitrogen dose per unit increases the yield by 49.96 kg/ha for the measured data, while it was 9.98 kg/ha for the simulated data. The model accurately estimated the yield of the N120 dose in 2020, but examining the other doses showed different results, maintaining the increasing trend (Figure 6).

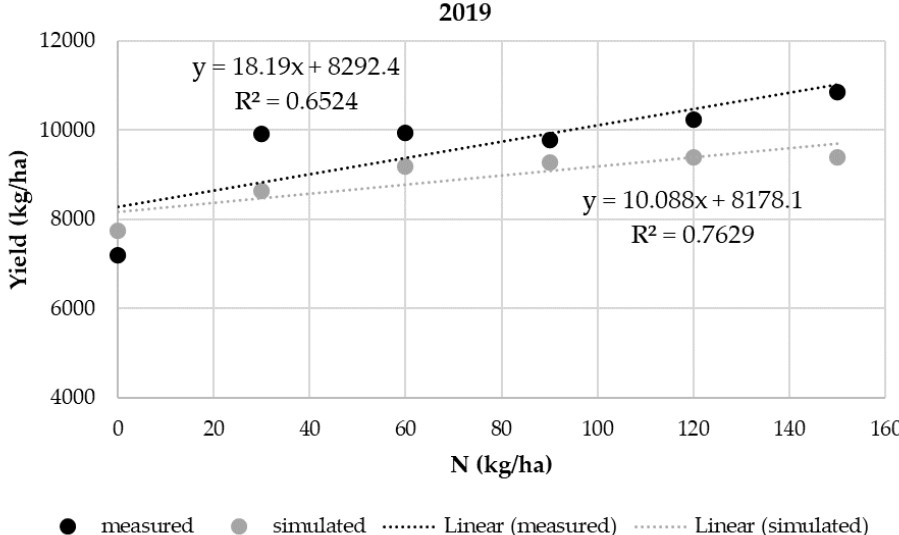

**Figure 5.** Comparisons between simulated and measured maize grain yield (2019).

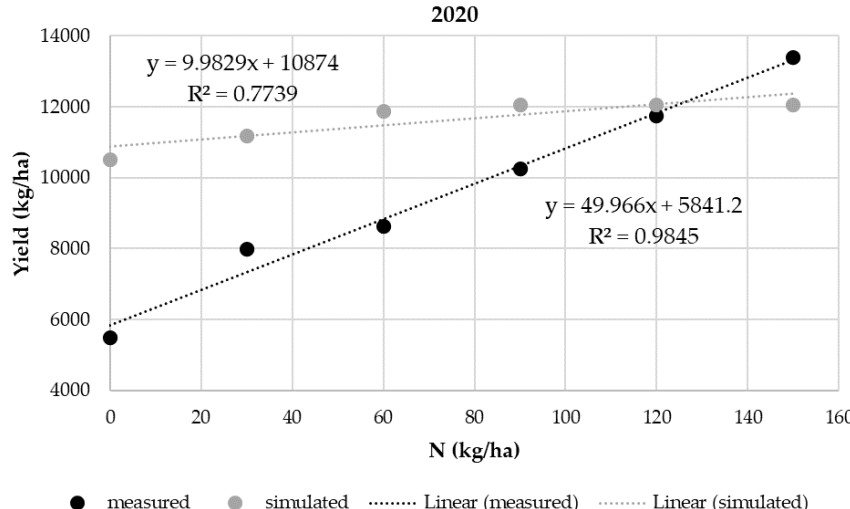

**Figure 6.** Comparisons between simulated and measured maize grain yield (2020).

According to similar studies [34–38], the CERES-Maize model was very sensitive to changes in climate factors. In summary, the model simulated the maize yields inaccurately in different N treatments and for the different growing season.

### 3.2. Results of the Statistical Analysis

According the analysis variance had a significant effect on years (2016–2020) and nitrogen doses. N fertilizer had significant impact on yield but the simulated data and measurement data do not show any significance together (Table 9). The Tukey analysis showed that there was a significant difference between the examined years. There was no difference between 2016 and 2020 (group A), in case of 2017, 2018, and 2019, but there was a significant difference compared to 2016 and 2020 (group B) (Table 10). On average, over the examined years, the Tukey grouping analysis showed that there was a significant difference between the various nitrogen doses. There were similar values in the case of the 150 and 120 N doses, while the second group involved the 90 and 60 N doses, and the third group included the 30 N dose and the fourth group is 0 nitrogen (Table 11). Based on the Tukey grouping analysis, there was no significant difference between measured and simulated values in the case of averaged years (Table 12).

**Table 9.** Component variance analysis on parameters.

| Source | DF | Adj SS | Adj MS | F-Value | *p*-Value |
|---|---|---|---|---|---|
| year | 4 | 57283327 | 14320832 | 5.78 | 0.001 |
| N | 5 | 114917792 | 22983558 | 9.28 | <0.001 |
| type | 1 | 2040570 | 2040570 | 0.82 | 0.396 |
| Error | 49 | 121361829 | 2476772 | | |
| Total | 59 | 295603519 | | | |

**Table 10.** Tukey Pairwise Comparisons on years.

| Year | N | Mean | Grouping | |
|---|---|---|---|---|
| 2016 | 12 | 11,626.6 | A | |
| 2020 | 12 | 10,605.8 | A | B |
| 2018 | 12 | 9484.2 | | B |
| 2019 | 12 | 9295.7 | | B |
| 2017 | 12 | 9000.7 | | B |

Means that do not share a letter are significantly different.

**Table 11.** Tukey Pairwise Comparisons on N fertilizer.

| N | N | Mean | Grouping | | | |
|---|---|---|---|---|---|---|
| 150 | 10 | 11,690.6 | A | | | |
| 120 | 10 | 11,264.3 | A | | | |
| 90 | 10 | 10,457.4 | A | B | | |
| 60 | 10 | 9954.2 | A | B | | |
| 30 | 10 | 9102.0 | | B | | C |
| 0 | 10 | 7546.9 | | | | C |

Means that do not share a letter are significantly different.

**Table 12.** Tukey Pairwise Comparisons on modeling.

| Type | N | Mean | Grouping |
|---|---|---|---|
| M | 30 | 10,187.0 | A |
| S | 30 | 9818.2 | A |

Means that do not share a letter are significantly different.

The analysis of variance had a significant effect on nitrogen, measured and simulated values. The Tukey grouping analysis showed that there was a significant difference between measured and predicted yields in 2016. The analysis of variance showed no significant effect on nitrogen and type. The Tukey grouping showed no significant difference between measured and simulated data in 2017. The analysis of variance showed no significant effect on nitrogen; however, the analysis of variance showed a significant effect on type factor. The Tukey grouping analysis showed significant difference between measured and simulated values in 2018. The analysis of variance had a significant effect on nitrogen; however, no significant effect was shown on measured and simulated values. The Tukey grouping analysis showed that there were not significantly difference between measured and simulated values in 2019. The analysis of variance did not have any significant effect on the examined parameters. The Tukey grouping analysis showed no significant difference between simulated and measured values in 2020 (Tables 13 and 14).

**Table 13.** Simple variance analysis on yield.

| Year | Source | DF | Adj SS | Adj MS | F-Value | *p*-Value |
|---|---|---|---|---|---|---|
| | N | 5 | 30900853 | 6180171 | 47.16 | <0.001 |
| 2016 | type | 1 | 1840050 | 1840050 | 14.04 | 0.013 |
| | Error | 5 | 655247 | 131049 | | |
| | Total | 11 | 33396151 | | | |
| | N | 5 | 26086128 | 5217226 | 2.86 | 0.137 |
| 2017 | type | 1 | 635720 | 635720 | 0.35 | 0.581 |
| | Error | 5 | 9125963 | 1825193 | | |
| | Total | 11 | 35847811 | | | |
| | N | 5 | 27856557 | 5571311 | 1.63 | 0.303 |
| 2018 | type | 1 | 58080000 | 58080000 | 16.98 | 0.009 |
| | Error | 5 | 17102215 | 3420443 | | |
| | Total | 11 | 103038772 | | | |
| | N | 5 | 8826853 | 1765371 | 6.99 | 0.026 |
| 2019 | type | 1 | 1563852 | 1563852 | 6.19 | 0.055 |
| | Error | 5 | 1262346 | 252469 | | |
| | Total | 11 | 11653051 | | | |
| | N | 5 | 28850326 | 5770065 | 2.20 | 0.204 |
| 2020 | type | 1 | 12415536 | 12415536 | 4.73 | 0.082 |
| | Error | 5 | 13118546 | 2623709 | | |
| | Total | 11 | 54384408 | | | |

**Table 14.** Tukey Pairwise Comparisons.

| Year | Type | N | Mean | Grouping | |
|------|------|---|------|----------|---|
| 2016 | S | 6 | 12,018.2 | A | |
| | M | 6 | 11,235.0 | | B |
| 2017 | S | 6 | 9230.83 | A | |
| | M | 6 | 8770.50 | A | |
| 2018 | M | 6 | 11,684.2 | A | |
| | S | 6 | 7284.2 | | B |
| 2019 | M | 6 | 9656.67 | A | |
| | S | 6 | 8934.67 | A | |
| 2020 | S | 6 | 11,623.0 | A | |
| | M | 6 | 9588.7 | A | |

## 4. Conclusions

The CERES-Maize model was evaluated using a long-term experiment in Hungary. In summary, simulated maize yields were not associated with site-specific measured maize yields from experimental plots. Our long-term experiment indicated that, as a result of increasing fertilization, crop yields increase. In addition to fertilization, yields were also affected by the weather. The model did not simulate the annual Sushi yields precisely. The examined years in this study differed significantly depending on the seasonal conditions. In the rainy year (2016), the hybrid Sushi reached a yield of 13.858 kg ha$^{-1}$, while the obtained yield was 3 tons less (10,860 kg ha$^{-1}$) even in the drought year (2019).

The simulated results of the model followed the increase in yields with increasing N dose. The measured and predicted yield during the years tracked the maize yields reasonably well for the 0N treatments. On specific treatment levels, the model accurately estimated yields for the Sushi hybrid, but in several cases, the model under- or overestimated yields. According the performed variance analysis, a significant effect was observed on crop year (2016–2020) and nitrogen doses. The Tukey analysis showed significant difference between each year. Even though the DSSAT-CERES Maize have showed some uncertainties associated in estimating the yields of different years, the increase in yield under the nitrogen dose was accurately modeled. The results of this study support the potential of using the model for the application of appropriate agrotechnics, including the determination of the nitrogen dose. Higher fertilizer doses resulted in higher yields each year.

In order to predict yields that meet the requirements of precision farming, care must be taken to collect data from experiments performed under optimal conditions. In addition, the data required for calibration should be collected from a location where detailed soil data are available. Analysis of varieties (hybrids) will help farmers to have more accurate decision support tools. Precision farming, which takes into account low spatial resolution management units, requires accurate, reliable crop yield models. Without validating these decision support systems for our growing conditions, we will not be able to make good agronomic decisions.

**Author Contributions:** A.Z.: data collection, modelling, and writing, A.S.: data curation, statistical analysis, and writing, J.N.: project administration and supervision, A.N.: conceptualization and writing. All authors have read and agreed to the published version of the manuscript.

**Funding:** The publication was supported by Széchenyi István Egyetem.

**Data Availability Statement:** Not applicable.

**Acknowledgments:** Project no. TKP2021-NKTA-32 was implemented with the support provided from the National Research, Development and Innovation Fund of Hungary, financed under the TKP2021-NKTA funding scheme. The authors thank the "Thematic Excellence Program—National Challenges Subprogram—Complex Precision Crop Production Research at Széchenyi István University (TKP2020-NKA-14)" project.

**Conflicts of Interest:** The authors declare no conflict of interest.

## Appendix A

**Table A1.** The physical and chemical parameters of the soil in the experimental area in 2017**.**

| | | | | | | | | | |
|---|---|---|---|---|---|---|---|---|---|
| **2017** | | | | | | | | | |
| **Layer Depth, cm** | **Organic Carbon %** | **Total Nitrogen %** | **pH in Water** | **Lower Limit, cm³ cm⁻³** | **Drained Upper Limit, cm³ cm⁻³** | **Saturated Water Holding Capacity, cm³ cm⁻³** | **Bulk Density g/cm³** | **Sat. Hydraulic Conduct, cm/h** | **Root Growth Factor, 0.0 to 1.0** |
| 5 | 1.58 | 0.16 | 7.3 | 0.21 | 0.424 | 0.495 | 1.24 | 0.15 | 1.000 |
| 10 | 1.58 | 0.16 | 7.3 | 0.21 | 0.424 | 0.495 | 1.24 | 0.15 | 1.000 |
| 15 | 1.58 | 0.16 | 7.3 | 0.21 | 0.424 | 0.495 | 1.24 | 0.15 | 1.000 |
| 20 | 1.58 | 0.16 | 7.3 | 0.21 | 0.424 | 0.495 | 1.24 | 0.15 | 1.000 |
| 25 | 1.34 | 0.13 | 7.2 | 0.202 | 0.41 | 0.489 | 1.26 | 0.15 | 0.638 |
| 30 | 1.34 | 0.13 | 7.2 | 0.202 | 0.41 | 0.489 | 1.26 | 0.15 | 0.577 |
| 35 | 1.34 | 0.13 | 7.2 | 0.202 | 0.41 | 0.489 | 1.26 | 0.15 | 0.522 |
| 40 | 1.34 | 0.13 | 7.2 | 0.202 | 0.41 | 0.489 | 1.26 | 0.15 | 0.472 |
| 45 | 0.97 | 0.1 | 7.2 | 0.192 | 0.391 | 0.477 | 1.3 | 0.15 | 0.427 |
| 50 | 0.97 | 0.1 | 7.2 | 0.177 | 0.377 | 0.484 | 1.28 | 0.68 | 0.387 |
| 55 | 0.97 | 0.1 | 7.2 | 0.177 | 0.377 | 0.484 | 1.28 | 0.68 | 0.35 |
| 60 | 0.97 | 0.1 | 7.2 | 0.177 | 0.377 | 0.484 | 1.28 | 0.68 | 0.317 |
| 65 | 0.6 | 0.06 | 8.0 | 0.167 | 0.358 | 0.472 | 1.32 | 0.68 | 0.287 |
| 70 | 0.6 | 0.06 | 8.0 | 0.18 | 0.367 | 0.472 | 1.32 | 0.15 | 0.259 |
| 75 | 0.6 | 0.06 | 8.0 | 0.18 | 0.367 | 0.472 | 1.32 | 0.15 | 0.235 |
| 80 | 0.6 | 0.06 | 8.0 | 0.18 | 0.367 | 0.472 | 1.32 | 0.15 | 0.212 |
| 85 | 0.5 | 0.05 | 8.4 | 0.178 | 0.362 | 0.47 | 1.33 | 0.15 | 0.192 |
| 90 | 0.5 | 0.05 | 8.4 | 0.178 | 0.362 | 0.47 | 1.33 | 0.15 | 0.174 |

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
