# Peer review of "Using the CERES-Maize Model to Simulate Crop Yield in a Long-Term Field Experiment in Hungary"

_agronomy, doi:10.3390/agronomy12040785_

Round 1

Reviewer 1 Report

The author used DSSAT-CERES Maize model and estimated 27 yields data in several cases with the major objectives (i) to access the ability of the Ceres-Maize model for predicting yields in long-term 14 experiments in Hungary (ii) to use the model to assess the effects of different nutrient management 15 (different nitrogen rates – 0, 30, 60, 90, 120, 150 kg ha-1 ). A long-term experiment with various N- 16 fertilizer applications, conducted in Látókép (Hungary), provided a reliable database to predict 17 maize yield under different conditions. This study seems very attractive and effective to measure the current requirements. The following major deficiencies notices as below:

  1. The author mistakenly wrote line 57-62 with out considering English grammar.
  2. Figure-2 graphics needs to improve
  3. The author used very old citation such as reference No. 4,6,7,11,15,16,17 and so on.
  4. The English language is very poor and not understandable

Author Response

Dear Reviewer,

Thank you for your comments concerning our manuscript entitled “Using the CERES-Maize Model to Simulate Crop Yield in a Long-Term Field Experiment in Hungary” (manuscript ID: agronomy- 1617904). Your comments are all valuable and very helpful for improving our paper. We have studied the comments carefully and made corresponding revisions. The revised portions are marked in the paper. Please verify the updated content in the manuscript.

We really appreciate your kindness and help to improve the quality of the manuscript.

Sincerely,

Authors

Reviewer 2 Report

THE PAPER  titled

 Using the DSSAT-CERES-Maize Model to Simulate Crop Yield 2 in a Long-Term Field Experiment in Hungary addressed very interesting topic. However, the current version need minor corrections as the following

  1. The authors have to highlight the new contribution here in a comparison with previous studies
  2. The authors have to highlight the importance of such study in the Crop Yield and applicability for the large scale
  3. In the discussion more explanation are required since the current version looks like a report
  4. The conclusion should have major findings in this study

Author Response

Dear Reviewer,

Thank you for your comments concerning our manuscript entitled “Using the CERES-Maize Model to Simulate Crop Yield in a Long-Term Field Experiment in Hungary” (manuscript ID: agronomy- 1617904). Your comments are all valuable and very helpful for improving our paper. We have studied the comments carefully and made corresponding revisions. Please, verify the updated content in the manuscript.

We really appreciate your kindness and help to improve the quality of the manuscript.

Sincerely,

Authors

Reviewer 3 Report

I could not find any novelty in this work, as many studies have already used  DSSAT-CERES-Maize Model for such purposes. For instance one of the related papers published on this model has been given here;

https://link.springer.com/article/10.1007/s10705-010-9396-y 

Author Response

Dear Reviewer,

Thank you for the comment about our manuscript “Using the CERES-Maize Model to Simulate Crop Yield in a Long-Term Field Experiment in Hungary” (manuscript ID: agronomy- 1617904).

I can honestly say, that I completely agree with you. There have been countless articles like this, and we have quoted some of them in manuscript - without being exhaustive.

The submission to the special issue was not for reason at all. Namely, we have tried to point out that these models are in most cases not suitable to handle high and fine accuracy (spatial and temporal) data. They are thus not able to meet the needs of precision crop production.

However, long term experiment databases can provide a very good basis to validate the models, and developers can use the databases.

Furthermore, I would like to highlight that if we do not use up-to-date data from long-term experiements (despite data from 40-50 years ago), how can we extend those conclusions to current challenges? We are not modellers, we are end-users with big data and precision ag. technologies and would like to built up more applicable agronomical relationships.

Anikó Nyéki

Round 2

Reviewer 1 Report

It is appreciated the authors efforts and simulating the DSSAT-CERES-Maize Model for Crop Yield in a Long-Term Field Experiment in Hungary. The overall experiments details shared by authors are appreciated, the following few points may considered to improve the manuscript quality.

  1. The citation of different articles in the second and third paragraphs are irrelevant.
  2. Figures 2 is not visible with specific coordinates, it is suggested to redraw and improve the graphics of all figures.
  3. Some grammatical mistakes found in articles s
  4. The author should brief the relevant section to describe the figures.

Author Response

Dear Reviewer,

Thank you for your comments concerning our manuscript entitled “Using the CERES-Maize Model to Simulate Crop Yield in a Long-Term Field Experiment in Hungary” (manuscript ID: agronomy- 1617904).

Sincerely,

Authors

The main corrections regarding the paper and the responses to your comments are as follows:

Comment: The citation of different articles in the second and third paragraphs are irrelevant.

Response: Thank you for the suggestion. We have deleted these citations and added to another section.

Comment: Figures 2 is not visible with specific coordinates, it is suggested to redraw and improve the graphics of all figures.

Some grammatical mistakes found in articles. The author should brief the relevant section to describe the figures.  

Response: We have tried to improve the figures (Figure 2-6) and the text of our manuscript with the descriptions.
